# Correlation of Health-Related Quality of Life with Negative Symptoms Assessed with the Self-Evaluation of Negative Symptoms Scale (SNS) and Cognitive Deficits in Schizophrenia: A Cross-Sectional Study in Routine Psychiatric Care

**DOI:** 10.3390/jcm12030901

**Published:** 2023-01-23

**Authors:** Jonas Montvidas, Virginija Adomaitienė, Darius Leskauskas, Sonia Dollfus

**Affiliations:** 1Psychiatry Department, Lithuanian University of Health Sciences, LT-50162 Kaunas, Lithuania; 2Psychiatry Department, University of Caen Normandy, 14000 Caen, France

**Keywords:** schizophrenia, negative symptoms, cognitive deficits, self-evaluation of negative symptoms, health-related quality of life

## Abstract

(1) Background: Schizophrenia is a severe mental disorder characterized by various symptom groups that tremendously affect health-related quality of life (HRQoL). We aimed to specify whether negative symptoms and cognitive deficits of schizophrenia correlate and can predict HRQoL. (2) Methods: Patients diagnosed with paranoid schizophrenia were invited to participate in the study. Participants were evaluated using the Montreal Cognitive Assessment (MoCA) and the Brief Psychiatric Rating Scale (BPRS) and were asked to fill out the Self-evaluation of Negative Symptoms scale (SNS) and the Medical Outcomes Short Form Survey (SF-36). Pearson’s and Spearman’s correlations were used to calculate the correlations between cognitive deficits and negative symptoms. We performed the receiver operating characteristic (ROC) analysis for the variables correlated with SF-36 scores. (3) Results: HRQoL correlated significantly with the negative symptoms; however, it did not correlate with cognitive deficits. ROC analysis showed that the abulia subscore of the SNS showed the most significant predictive potential of HRQoL. (4) Conclusions: Negative symptoms correlate more significantly with the HRQoL than cognitive symptoms. The SNS offers the possibility of predicting the HRQoL of patients with schizophrenia and is useful as a screening tool in clinical practice.

## 1. Introduction

Schizophrenia is a debilitating mental disorder characterized by an insidious course with relapsing-remitting positive symptoms (hallucinations, delusions, disorganized thought process), chronically present negative symptoms (blunted affect, anhedonia, alogia, avolition, social withdrawal), and cognitive deficits (attention, speed of processing, memory, working memory, reasoning, and problem-solving, as well as social cognition domains, such as emotion processing and theory of mind) [1]. According to data published by the World Health Organization, schizophrenia affects approximately one person out of every 300 and mostly starts at a young age [2].

Around 90 percent of patients diagnosed with schizophrenia have cognitive deficits, and approximately 50 to 60 percent have negative symptoms [3,4]. Both cognitive deficits and negative symptoms have predictive value for the onset of psychosis and the severity and the outcomes of the disease [5,6,7]. Moreover, negative symptoms are present in 50–70% of first-episode psychosis patients later diagnosed with schizophrenia, and around 10 to 30% of them develop at least one persistent negative symptom [8,9]. Cognitive deficits and negative symptoms remain chronically present during the illness [10]. Neither cognitive deficits nor negative symptoms significantly correlate with the severity of positive symptoms or the duration of untreated psychosis [11,12].

Schizophrenia significantly impacts the objective social, occupational, and everyday functioning and the subjective Health-Related Quality of Life (HRQoL) of schizophrenia patients and their caretakers [13]. Objective everyday functioning refers to how well a person performs various tasks. Subjective HRQoL refers to how health influences the perceived well-being in 3 domains of life: physical, mental, and social [14,15]. HRQoL has been found to be an independent predictor of relapse in schizophrenia [16]. A phenomenon called the “insight paradox” was described that greater insight in schizophrenia was negatively correlated with the subjective HRQoL [17]. Among the 3 domains of HRQoL, the social domain was reported to have the lowest score [18]. Therefore, assessing HRQoL in schizophrenia is necessary to understand how this disorder impacts the life of the patients and to set the main treatment targets to minimize the impact of this mental disorder [18].

Most studies that evaluated negative symptoms of schizophrenia used extensive and challenging to use questionnaires. The most widely used instruments for negative symptoms assessment were the Positive and Negative Symptoms Scale (PANSS) for the Assessment of Negative Symptoms (SANS) [19]. These scales take at least 45 min to complete, and PANSS needs special training to be administered and scored correctly, which significantly impedes its use in daily bedside practice [20,21,22,23]. Moreover, PANSS and SANS have proven content validity problems [24]. The European Psychiatrists Association (EPA) Guidance on assessing negative symptoms advocates the so-called ‘second generation’ scales. It encourages using self-assessment tools such as the Self-assessment scale of Negative Symptoms (SNS) [19].

Most research articles about the relationship between cognitive deficits and functioning or HRQoL employed expensive, corporation-owned, complicated questionnaires that took a long time to complete. The most often used tools were Cambridge Neuropsychological Test Automated Battery (CANTAB), CogState, and Repeatable Battery for the Assessment of Neuropsychological Status (BRANS) [25]. The NIMH-Measurement and Treatment Research to Improve Cognition in Schizophrenia (MATRICS) Consensus Cognitive Battery (MCCB) is a diagnostic tool that is recommended by the EPA to be used for cognitive function assessment for patients with schizophrenia [26]. However, MCCB takes at least 60–90 min to complete, making it difficult to use in a bedside practice [27,28]. Moreover, none of the tools recommended for cognitive symptoms assessment by the EPA are available in Lithuanian.

Recognizing the need for further research on the relationship between HRQoL and negative symptoms and cognitive deficits of schizophrenia conducted using tools applicable in daily practice, we aimed to evaluate negative symptoms and cognitive deficits of schizophrenia with easily administered scales and to assess the correlations between negative symptoms, cognitive deficits, and subjective health-related quality of life.

## 2. Materials and Methods

### 2.1. Participants

Patients of the Lithuanian University of Health Sciences Hospital Kaunas Clinics Psychiatry department diagnosed with paranoid schizophrenia (according to the ICD-10 AM diagnostic criteria, diagnostic code F20.0) were invited to participate in the study during routine clinical care. The period of data collection was between the beginning of March 2019 and January 2020. The authors of this study worked as psychiatrists in the inpatient and outpatient psychiatric wards of the study center. The authors of this study asked other psychiatrists to inform them about patients with schizophrenia without active psychotic symptoms. Patients on the last day of their hospitalization and without active psychotic symptoms were informed about our study and invited to join. Outpatients that visited their psychiatrists for a routine visit and were evaluated by the treating psychiatrist as presenting no active psychotic symptoms were informed about the study and were asked to join. One of the researchers screened participants who were asked to join our study for inclusion and exclusion criteria. Inclusion criteria were diagnosis of schizophrenia with no acute psychotic symptoms present during the evaluation and aged 18–65. Exclusion criteria were intellectual disability, addiction to psychoactive substances, and acute psychotic symptoms during the primary assessment. We screened the patients for active psychotic symptoms with the Lithuanian version of the Mini International Neuropsychiatric Interview (MINI) module L, “Psychotic disorders.” We wanted to test the relationship between negative symptoms, cognitive deficits and the subjective HRQoL; therefore, we controlled for psychosis and did not include patients with active, positive symptoms that can induce secondary negative symptoms. Depression was evaluated using the Brief Psychiatric Rating scale-18 item 9 “Depression”. We did not control for the number of medications, the dosage, and the duration of treatment. Participants deemed appropriate to be included in the study were asked to sign an informed consent form. Around 150 patients with schizophrenia were treated in the study center during the period of data collection; out of them, around 100 were recommended to our team and were screened, and 67 met the inclusion criteria and signed the informed consent form to participate in the study. We did not collect the data of screened patients.

### 2.2. Bioethics

Bioethics approval No. BE-2-22 was received from the Kaunas Regional Biomedical Research Ethics Committee on 1 March 2019.

### 2.3. Procedure and Measures

After the participants signed the informed consent form, they were included in the study. Participants were screened during routine clinical care; therefore, short and quick evaluation tools were selected—the Montreal Cognitive Assessment (MoCA) and the Brief Psychiatric Rating Scale (BPRS). After screening, they were asked to complete the Medical Outcomes Short Form Survey (SF-36) and the Self-evaluation of Negative Symptoms Scale (SNS). Sociodemographic data regarding the sex, age, and years of education of participants were collected. Patients were also asked to rate their general well-being from 1 (feeling horrible) to 10 (feeling wonderful).

The MoCA is a screening tool developed to detect mild cognitive impairments. It evaluates short-term memory, visuospatial abilities, executive functions, attention, language, and orientation. It has a highest score of 30 points and takes 10–15 min to complete. The MoCA evaluates visuospatial/executive (MoCA-VE), naming (MoCA-N), attention (MoCA-A), language (MoCA-L), abstraction (MoCA-AB), delayed recall (MoCA-DR), and orientation (MoCA-O) subscores and the total score (MoCA-TS) [29]. The MoCA is considered sensitive to detect mild and severe cognitive impairment in patients with schizophrenia, therefore validating it as an appropriate cognitive deficit screening tool for patients with schizophrenia. It has been validated in a schizophrenia patient sample that scoring below 26 points indicates a mild cognitive impairment, and scoring below 23 points indicates severe cognitive impairment [30,31,32]. Even though the MoCA is not a tool that is widely accepted to use for the assessment of cognitive symptoms, it is the only alternative to the Mini-Mental Examination Scale (MMSE) available in Lithuania.

The BPRS is one of the most widely used psychiatric rating scales. We used a version of the BPRS consisting of 18 items evaluating symptoms of depression, anxiety, agitation, and psychosis, as well as negative symptoms. Every item is scored from 1 (not present) to 7 (extremely severe) [33]. The scoring is based on the clinical interview and the patient’s behavior. Depressive symptoms are evaluated with the ninth item, “Depressive mood” (BPRS-D) [34,35]. Most researchers employed the negative symptoms of schizophrenia subscores (BPRS-N) consisting of a sum of the scores of items 3, “Emotional withdrawal,” 13, “Motor retardation,” and 16, “Blunted affect” [36].

The SNS is a ‘second generation’ subjective negative symptoms evaluation tool that evaluates all five domains of negative symptoms of schizophrenia [37]. It is recommended in the current EPA Guidance paper on negative symptom assessment [19]. A patient is asked to read the 20 statements listed in the SNS and then mark whether they agree (2 points), mildly agree (1 point), or disagree (0 points) with each of the statements. The maximum score is 40 points, with each subdomain of negative symptoms having a top score of 8 points. Questions 1 to 4 evaluate social withdrawal (SNS-SW), 5 to 8 evaluate reduced emotional range (SNS-RER), 9 to 12 evaluate alogia (SNS-A), 13 to 16 evaluate avolition (SNS-AV), and questions 17 to 20 assess anhedonia (SNS-AN). We can add the five subscores to obtain a total score (SNS-TS) [37]. It only takes around 5 min to complete and score the SNS. The Lithuanian version of SNS has been validated and shown good psychometric properties [38].

The SF-36 is one of the most widely used HRQoL evaluation tools. It is free and easy to self-administer and takes about 7–10 min [39]. The SF-36 measures eight HRQoL aspects: physical functioning (SF-36-PF), physical role-functioning (SF-36-RP), bodily pain (SF-36-BP), general health (SF-36-GH), vitality (SF-36-VT), social functioning (SF-36-SF), role emotional (SF-36-RE), and mental health (SF-36-MH). Component analyses showed that there are two distinct concepts measured by the SF-36: a physical dimension, represented by the Physical Component Summary (PCS), and a mental dimension, represented by the Mental Component Summary (MCS) [40]. Every HRQoL aspect is assessed by a score varying from 0 to 100, with a higher score meaning better HRQoL. These subscales are not disease or treatment specific. The SF-36 subscales were found suitable to administer, and the scores were reliable for patients with schizophrenia [41].

### 2.4. Statistical Analysis

We used the Chi-square test to see if there were statistically significant differences in sample distribution according to different variables. Internal consistency was calculated for the BPRS-N, MoCA-TS, and SNS-TS. The means of BPRS-D and BPRS-N scores, five SNS subscores and the SNS-TS, MoCA subscores, and MoCA-TS and SF-36 scores were calculated. We grouped the sample into groups according to the SNS total score (≤20 or >20), MoCA total score (≥26 or <26), and SF-36 scores (<50 or ≥50) in order to evaluate for demographic differences between different patient groups. We chose the middle of the score range for SNS and SF-36 because we did not find data regarding cut-off scores between severe and mild negative symptoms and good and bad HRQoL. All scores and subscores were tested for normality of distribution using the Shapiro–Wilk test. Student t-test was used for normally distributed and parametric variables, and the Mann–Whitney test was used for the not normally distributed and nonparametric variables. Our main null hypothesis was that the subjective HRQoL did not correlate with negative symptoms and/or cognitive deficits of schizophrenia. Pearson’s correlation was used to calculate the correlations between normally distributed variables, and Spearman’s correlation was used for variables that were not distributed normally. We performed the receiver operating characteristic (ROC) analysis for the variables correlated with SF-36 scores. We set SF-36 <50 as an expected outcome. An area under the curve (AUC) of 0.5 would tell us that our test was not able to distinguish the true positives (where the SF-36 score is actually <50) and false positives (where SF-36 is >50 even though we expected it to be ≥50). An AUC of 1 would tell us that all of the positives are true positives. An excellent test would be an AUC of 0.9 or more, a good one would be an AUC of 0.8 or more, and a fair one would be an AUC of 0.7 or more.

The level of significance was kept at 95% (*p* < 0.05). A total of 56 correlation analyses were performed; therefore, the Bonferroni correction for *p*-values was 0.018. We used the Statistical Package for the Social Sciences version 27 for the statistical analysis.

## 3. Results

### 3.1. Demographic Data

The sample consisted of 67 respondents. It included significantly (*p* = 0.02) more females (*n* = 43, 64.2%) than males (*n* = 24, 35.8%). The mean age was 41.51 (SD 13.76, CI 95% 38.15–44.86). Age was not normally distributed (*p* = 0.005). There were no age differences between the sexes (z = −1.564, *p* = 0.118). The mean of years of education was 14.9 (SD 3.34, CD95% 14.08–15.71). Years of education were not normally distributed (*p* = 0.024) and did not differ between sexes (z = −0.251, *p* = 0.802). The mean score of general well-being was 6.27 (SD 1.871, CD95% 5.81–6.73) and was not normally distributed (*p* = 0.005). The general well-being score did not differ between the sexes (z = −0.967, *p* = 0.334).

There were no significant differences in the number of participants between SNS-TS ≤20 or >20 groups (*p* = 0.282). There was a significant difference between respondent count in MoCA-TS ≥26 (*n* = 20; 29.9%) and <26 (*n* = 47; 70.1%) groups (*p* < 0.001). There were statistically significant differences in SF-36 <50 and ≥50 groups in SF-36-PF (12 vs. 55, *p* <0.001), SF-36-RP (20 vs. 47, *p* <0.001), and SF-36-BP (54 vs. 13, *p* < 0.001).

### 3.2. Internal Consistency, Mean Scores and Mean Ranks

Cronbach’s alpha of the BPRS-N (α = 0.857), SNS-TS (α = 0.82), and the five subscores (α = 0.76), MoCA-TS (α = 0.769) and the SF-36 (α = 0.858) showed good internal consistency.

The mean scores of the BPRS-D, BPRS-N, SNS subscores, SNS-TS, MoCA subscores, and MoCA-TS and SF-36 scores are given in Table 1. None of the scores differed significantly between the sexes.

### 3.3. Correlations

None of the scores correlated with age, except for MoCA-N (rho = −0.364, *p* = 0.002) and MoCA-TS (rho = −0.345, *p* = 0.004). None of the scores correlated with years of education. None of the MoCA or BRPS-D scores correlated with the general well-being score. However, all five SNS subscores, SNS-TS and BPRS-N, correlated significantly with the general well-being score. Most of the SF-36 scores correlated significantly with the general well-being score. Correlations with the general well-being score are provided in Table 2.

The MoCA did not correlate with BPRS-N, BPRS-D, and SNS scores, except for MoCA-L with BPRS-N (rho = −0.3, *p* = 0.014), MoCA-AB with BPRS-N (rho = −0.349, *p* = 0.004), MoCA-TS with SNS-A (rho = −0.243, *p* = 0.048), MoCA-VE with SNS-AN (rho = −0.408, *p* < 0.001) and MoCA-TS with SNS-AN (rho = −0.319, *p* = 0.008). The MoCA did not correlate with SF-36 scores. The BPRS-D had only one statistically significant correlation with SF-36-PF (rho = −0.324, *p* = 0.008).

We found that SF-36-PF correlated with BPRS-N, SNS-TS, and every SNS subscore except for SNS-BA. SF-36-RP did not correlate with any of the SNS scores or BPRS-N. SF-36-BP correlated with SNS-AV and SNS-AN. SF-36-RE correlated with SNS-AV. SF-36-GH, SF-36-VT, SF-36-SF, and SF-36-MH correlated with every SNS score and BPRS-N. SNS, BPRS, and SF-36 correlations are provided in Table 3.

### 3.4. Receiver Operating Characteristic (ROC) Analysis

Because MoCA scores did not correlate with SF-36 scores, we did not perform a ROC analysis for the MoCA. The ROC analysis for BPRS-D, BPRS-N, and SNS scores had mixed results. The ROC analysis with a statistically significant AUC ≥ 0.7 is given in Table 4.

No AUC ≥ 0.9 was found. Only SNS-AV with SF-36-GH (AUC = 0.823) and SNS-TS with SF-36-MH (AUC = 0.809) had AUC ≥ 0.8. We found that the SNS-AV score of 3.5 predicted that SF-36-GH would be less than 50, with a sensitivity of 79.5% and specificity of 75%. SNS-TS score of 20.5 predicted that SF-36-MH would be less than 50 with a sensitivity of 70% and specificity of 73%.

## 4. Discussion

We found that negative symptoms of schizophrenia had a stronger correlation with HRQoL than cognitive deficits when evaluated with quick bedside tools SNS, BPRS and MoCA. The correlation between cognitive deficits of schizophrenia and the HRQoL was insignificant. We also found that negative symptoms of schizophrenia, assessed with a short self-evaluation scale, were predictive of the HRQoL in schizophrenia.

The unique finding of our study is that the avolition subdomain of negative symptoms was the most predictive of reduced HRQoL. To our knowledge, this is the first study where ROC analysis was performed for HRQoL prediction using a scale for negative symptom assessment. Dollfus et al. performed a ROC analysis in a study with SNS and found that scoring 7 points or more on the SNS separated healthy controls and patients with schizophrenia with a sensitivity of 92.7% and specificity of 85.9% [42]. We believe that the findings of our ROC analysis of the SNS results complement the results of Dollfus et al. and further prove the validity of SNS as a screening tool regarding the prediction of poor life quality for patients with schizophrenia.

Our results of an insignificant correlation between the subjective HRQoL and cognitive deficits in schizophrenia are similar to the results of other authors, who found that cognitive deficits were closely linked to poorer everyday functioning but not subjective HRQoL. A relationship between cognitive functioning and real-life functioning has been found in a 5-year large-scale longitudinal study by Mucci et al. [43]. The EPA Guidance paper on the evaluation of cognitive deficits of schizophrenia described that cognitive deficits of schizophrenia are closely linked to everyday functioning but not to the subjective HRQoL [26]. A meta-analysis by Arielle et al. showed that cognitive deficits had a non-statistically significant relationship with self-reported HRQoL [44]. Various other authors also conclude that schizophrenia is characterized by deficits in various cognitive domains that have a more significant effect on objective functioning compared to negative or positive symptoms but have less influence on the subjective HRQoL [26,45,46]. Moreover, Domenech et al. found that patients with more significant cognitive deficits reported higher HRQoL on the SF-36 scale [47].

On the other hand, some authors find different results. For example, Apteinin et al. found that executive functions and working memory deficits were associated with lower self-reported quality of life [48]. Kurtz et al. found similar results in a 5-year follow-up study; however, they found that negative and positive symptoms but not cognitive deficits were independent predictors of objective psychosocial status. However, they managed to find a link between cognitive deficits and “life satisfaction” [49]. This could be explained by much greater sample sizes and different methodologies and warrants further research.

We found that negative symptoms correlated strongly and significantly with the HRQoL. Many other authors have reported similar findings. Greater PANSS negative symptoms scores were associated with worse SF-36 physical and mental scores [47,50]. Pukrop et al. found that the reduction of negative symptoms severity significantly improved SF-36 scores [41]. Rabinowitz et al. found that negative symptoms were more correlated to the reduction of QoL. However, a combination of prominent negative and prominent positive symptoms had the greatest correlation with reduced QoL [6].

On the other hand, Chou et al. found that psychosocial factors had the most significant effect on the HRQoL, and depressive symptoms had the most significant effect out of the psychopathological factors [46]. This might be explained by different assessment tools used to assess depressive and negative symptoms of schizophrenia. Chou et al. used PANSS, a hetero-assessment tool, to evaluate negative symptoms and a self-assessment scale (Beck Depression Inventory-II, BDI-II) to assess depressive symptoms, while we did the opposite and used hetero-assessment of depressive symptoms with the BPRS and used self-rating for negative symptoms with SNS. Chemerinski et al. found that using only specific few items within BDI-II and not the entire BDI-II provides a more clinically accurate assessment of depression in schizophrenia [51]. Additionally, other authors found that depressive and negative symptoms often overlap, and depressive symptoms often constitute secondary negative symptoms [52]. Even though Dollfus et al. found that SNS is an appropriate tool to screen negative symptoms regardless of the severity of depressive symptoms, we believe that further investigation into the ability of self-assessment tools to distinguish between secondary negative symptoms caused by depression and primary negative symptoms is required [42].

There is an ongoing discussion about what type of HRQoL measures are applicable for patients with schizophrenia. HRQoL measures can be divided into two groups: specific disease-targeted measures and generic measures. The disease-targeted measures are created to evaluate a particular disorder, whereas generic measures can be applied to any disorder. Generic measures can be further specified as profile-based (evaluating multiple aspects) or preference-based (producing a single score). We used the SF-36, which is a generic profile measure [14]. The SF-36 was proven applicable in schizophrenia research by some researchers [7,47]. However, other researchers found unclear results about the applicability of generic HRQoL measures for patients with schizophrenia [53,54]. Therefore, we recognize that using a generic HRQoL measure can be considered one of the limitations of our study.

Another limitation of our study is the use of the MoCA for the evaluation of cognitive deficits of schizophrenia because this scale is not among the tools that are recommended for the assessment of cognitive deficits [26]. However, we had to comply with the fact that only the MoCA and MMSE are currently available in Lithuania. The MoCA is more effective than the MMSE when evaluating cognitive deficits in a sample of patients with schizophrenia [55,56,57]. Moreover, Belvederi et al. have compared the MoCA to the Screen for Cognitive Impairment in Psychiatry (SCIP), which is recognized and recommended for the evaluation of cognitive deficits by the EPA. The MoCA showed similar, albeit worse, results compared to the SCIP [58]. Our research team is currently in the process of validating the Lithuanian version of the SCIP.

Other limitations are a relatively small sample size, broad inclusion criteria, not using a ‘second-generation’ rater-based assessment tools for negative symptom assessment, and using a short-form evaluation of depressive symptoms. A bigger sample size might have improved the normality of the distribution of the variables and increased the significance of some correlations between HRQoL and cognitive deficits. Making the inclusion criteria narrower and controlling for medications used, years of disease, and other criteria might have helped make our results more reliable. Using a ‘second-generation’ hetero-assessment tool for evaluating negative symptoms might have helped to increase the validity of our negative symptom assessment. However, the ability of patients with schizophrenia to self-evaluate was proven extensively [59,60,61]. Direct comparison of precision for prediction of the HRQoL of such tools as the Brief Negative Symptoms Scale and SNS would be recommended. Moreover, using a more detailed evaluation of depressive symptoms could have yielded a more pronounced correlation of depression symptoms with other symptom groups.

## 5. Conclusions

We may conclude that HRQoL correlated significantly with negative symptoms but did not correlate with cognitive deficits of patients diagnosed with schizophrenia. Negative symptoms evaluated with both ‘first generation’ observer-rated and ‘second generation’ self-assessment tools correlated significantly with HRQoL. A reduction of HRQoL can be predicted with SNS, especially the avolition subscore of SNS, which warrants further investigation of SNS as a screening tool for the quality of life of patients with schizophrenia.

## Figures and Tables

**Table 1 jcm-12-00901-t001:** Mean scores of SNS, BPRS, MoCA, and SF-36.

Scale	Mean Score	CI 95 Proc.
SNS-SW	4.33 (2.642)	3.68–4.97
SNS-RER	3.49 (1.691)	3.08–3.91
SNS-A	4.03 (2.933)	3.33–4.75
SNS-AV	4.03 (2.202)	3.49–4.57
SNS-AN	3.27 (2.1)	2.76–3.78
SNS-TS	18.61 (8.792)	16.47–20.76
BPRS-D	2.39 (1.255)	2.08–2.69
BPRS-N	10.42 (3.947)	9.46–11.38
MoCA-VE	3.18 (1.476)	2.82–3.54
MoCA-N	2.84 (0.51)	2.71–2.96
MoCA-A	4.36 (1.544)	3.98–4.73
MoCA-L	1.58 (0.781)	1.39–1.77
MoCA-AB	1.31 (0.763)	1.13–1.5
MoCA-DR	2.93 (1.418)	2.58–3.27
MoCA-O	5.81 (0.557)	5.67–5.91
MoCA-TS	22.82 (4.376)	21.75–23.89
SF-36-PF	75.75 (24.313	69.82–81.68
SF-36-RP	52.24 (20.749)	47.18–57.30
SF-36-BP	26.63 (26.863)	20.07–33.18
SF-36-GH	43.28 (20.954)	38.17–48.39
SF-36-VT	41.79 (21.047)	26.66–46.92
SF-36-SF	45.36 (24.742)	39.32–51.93
SF-36-RE	37.76 (37.159)	28.7–46.83
SF-36-MH	52.54 (18.94)	47.92–57.16

**Table 2 jcm-12-00901-t002:** Correlation of SNS-TS and subscores, BPRS-N and SF-36 subscores with the general well-being score.

Scale	Pearson r/Spearman Rho	*p*
SNS-SW	−0.459	<0.001
SNS-RER	−0.428	<0.001
SNS-A	−0.414	<0.001
SNS-AV	−0.592	<0.001
SNS-AN	−0.395	<0.001
SNS-TS	−0.593	<0.001
BPRS-N	−0.443	<0.001
SF-36-PF	0.497	<0.001
SF-36-RP	0.253	0.039
SF-36-BP	−0.159	0.2
SF-36-GH	0.585	<0.001
SF-36-VT	0.619	<0.001
SF-36-SF	0.407	<0.001
SF-36-RE	0.451	<0.001
SF-36-MH	0.478	<0.001

**Table 3 jcm-12-00901-t003:** SF-36, SNS, and BPRS-N correlations.

SF-36 Score	SNS/BPRS Score	Pearson r/Spearman Rho	*p*
SF-36-PF	SNS-TS	−0.531	<0.001
SF-36-RP	SNS-TS	−0.129	0.263
SF-36-BP	SNS-TS	0.288	0.018
SF-36-GH	SNS-TS	−0.589	<0.001
SF-36-VT	SNS-TS	−0.627	<0.001
SF-36-SF	SNS-TS	−0.496	<0.001
SF-36-RE	SNS-TS	−0.229	0.062
SF-36-MH	SNS-TS	−0.59	<0.001
SF-36-PF	SNS-SW	−0.427	<0.001
SF-36-RP	SNS-SW	−0.121	0.329
SF-36-BP	SNS-SW	0.172	0.164
SF-36-GH	SNS-SW	−0.38	0.002
SF-36-VT	SNS-SW	−0.49	<0.001
SF-36-SF	SNS-SW	−0.482	<0.001
SF-36-RE	SNS-SW	−0.198	0.109
SF-36-MH	SNS-SW	−0.51	<0.001
SF-36-PF	SNS-RER	−0.188	0.129
SF-36-RP	SNS-RER	0.053	0.667
SF-36-BP	SNS-RER	0.086	0.491
SF-36-GH	SNS-RER	−0.342	0.005
SF-36-VT	SNS-RER	−0.44	<0.001
SF-36-SF	SNS-RER	−0.394	<0.001
SF-36-RE	SNS-RER	−0.245	0.046
SF-36-MH	SNS-RER	−0.509	<0.001
SF-36-PF	SNS-A	−0.373	0.002
SF-36-RP	SNS-A	−0.085	0.492
SF-36-BP	SNS-A	0.218	0.077
SF-36-GH	SNS-A	−0.448	<0.001
SF-36-VT	SNS-A	−0.375	0.002
SF-36-SF	SNS-A	−0.413	<0.001
SF-36-RE	SNS-A	−0.177	0.153
SF-36-MH	SNS-A	−0.427	<0.001
SF-36-PF	SNS-AV	−0.518	<0.001
SF-36-RP	SNS-AV	−0.238	0.053
SF-36-BP	SNS-AV	0.323	0.008
SF-36-GH	SNS-AV	−0.568	<0.001
SF-36-VT	SNS-AV	−0.629	<0.001
SF-36-SF	SNS-AV	−0.329	0.007
SF-36-RE	SNS-AV	−0.368	0.002
SF-36-MH	SNS-AV	−0.458	<0.001
SF-36-PF	SNS-AN	−0.469	<0.001
SF-36-RP	SNS-AN	−0.146	0.238
SF-36-BP	SNS-AN	0.284	0.02
SF-36-GH	SNS-AN	−0.442	<0.001
SF-36-VT	SNS-AN	−0.388	0.001
SF-36-SF	SNS-AN	−0.374	0.002
SF-36-RE	SNS-AN	−0.163	0.188
SF-36-MH	SNS-AN	−0.467	<0.001
SF-36-PF	BPRS-N	−0.375	0.002
SF-36-RP	BPRS-N	−0.007	0.955
SF-36-BP	BPRS-N	0.175	0.157
SF-36-GH	BPRS-N	−0.415	<0.001
SF-36-VT	BPRS-N	−0.417	<0.001
SF-36-SF	BPRS-N	−0.354	0.003
SF-36-RE	BPRS-N	−0.155	0.209
SF-36-MH	BPRS-N	−0.486	<0.001

**Table 4 jcm-12-00901-t004:** The area under the curve (AUC) ≥ 0.7.

SF-36 Score	SNS/BPRS Score	AUC	*p*
SF-36-PF	SNS-AV	0.711	0.023
	BPRS-D	0.719	0.018
SF-36-RP	SNS-AV	0.709	0.007
SF-36-GH	SNS-A	0.729	0.001
	SNS-AV	0.823	<0.001
	SNS-TS	0.776	<0.001
SF-36-VT	SNS-AV	0.762	<0.001
	SNS-TS	0.749	0.001
SF-36-SF	SNS-A	0.728	0.002
SF-36-MH	BPRS-N	0.737	0.001
	SNS-SW	0.786	<0.001
	SNS-RER	0.759	<0.001
	SNS-A	0.706	0.004
	SNS-AV	0.743	0.001
	SNS-AN	0.745	0.001
	SNS-TS	0.809	<0.001

## Data Availability

Data are stored in the personal storage of the corresponding author and is not available online.

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
