# Peer review of "Correlation of Health-Related Quality of Life with Negative Symptoms Assessed with the Self-Evaluation of Negative Symptoms Scale (SNS) and Cognitive Deficits in Schizophrenia: A Cross-Sectional Study in Routine Psychiatric Care"

_jcm, 2023, doi:10.3390/jcm12030901_

Round 1
Reviewer 1 Report
The analysis done gave enough space about the aspect of the tool used that warrants mention in the title of the study.
Elaboration is needed as to why participants with positive symptoms were excluded and why they only focused on negative symptoms.
There is inadequate evaluation of depression that is not uncommon in the study population; how that was taken care of needs to be included in the manuscript.
The method of recruitment is unclear. Anything done for negative and cognitive symptoms come on the way of recruitment? How many enrolled and how many dropouts are not clear.
Author Response
Dear Reviewer,
Thank you for your informative and helpful comments about our article. We made some changes according to your insights, which I want to share here:
First, we included the self-evaluation of the negative symptoms scale in the title of our article, as you suggested. Therefore, now the title stands as “Correlation of health-related quality of life with negative symptoms assessed with the Self-evaluation of negative symptoms scale (SNS) and cognitive deficits in schizophrenia: a cross-sectional study in routine psychiatric care.”
We included a more detailed explanation of why patients with active psychotic symptoms were excluded from our study (see page 4 in the chapter ‘participants’). We wanted to evaluate the correlation between negative symptoms and cognitive deficits of schizophrenia with the subjective health-related quality of life. We thought active psychotic symptoms could distort these correlations since positive symptoms can induce secondary negative symptoms.
You mentioned that we did an inadequate evaluation of depression. We evaluated depressive symptoms with item nine (“depression”) of the Brief Psychiatric Rating Scale (BPRS). We have mentioned this in the method and result sections (see page 3 in the chapter ‘participants’, page 5 Table 1, and page 6 results subsection ‘correlations’). We agree that this is a limitation of our paper. Therefore we have expanded your limitations section about the limited evaluation of depression (see pages 9-10 ‘discussion’).
Moreover, we explained the recruitment process more in the methods section (see page 3, ‘participants’)—all of the authors of this study except prof. S. Dollfus worked in the Psychiatry ward of the Lithuanian University of Health Sciences, which was the study center for our researcher. The authors discussed the inpatients and outpatients with schizophrenia with other psychiatrists who worked in this Psychiatry department and asked them about their patients. Those patients who were deemed not to have active psychotic symptoms by their treating psychiatrist and were in the outpatient setting or on the last day of their inpatient setting were recommended to join our study. We screened the patients urged to join our study and asked those who fit the inclusion criteria to join. We collected data from only those patients who agreed to join the study and signed the informed consent form. We did not collect data about other patients.
In conclusion, your comments were highly insightful and helped us improve our paper. We hope that the changes we have made will satisfy you.
Reviewer 2 Report
The aim of this study was to examine whether negative symptoms and cognitive deficits of schizophrenia correlate and can predict the scores on HRQoL in clinical practice rather than in a research setting. While this may be a laudable aim for the non-research clinician, it may not really add to the body of knowledge on the HRAoL. The authors spend the introduction in a rather negativistic tone about the current overly complicated assessment instruments, rather than citing the specific literature on the HRQoL. It would be helpful for them to review that literature and their stating what they intend to add to it. They state that “It has been accepted that treatment of schizophrenia should aim at improving the HRQoL…”. This reviewer has not seen this in any treatment guidelines as generally the goal of treatment is not to improve the score on a rating scale, but to get the patient better in a specific domain of functioning and then to demonstrate the improvement by a change in a corresponding rating scale.
Three areas are assessed at one time point which is not clearly stated in a mixed patient sample: Inpatients at the point of discharge and stable outpatients. They need to show that there were no clinical differences between these two patient groups. The assessment tools are indeed brief and not “state-of the art” as the authors state, but more driven by their shortness and practicality. However, the MoCa is clearly not an acceptable cognitive assessment tool for assessment of non-organic cognitive deficits in schizophrenia. They state that they included patients with schizophrenia, paranoid type, but then they do not indicate in the Inclusion criteria any subtypes. They should also provide information on medications, which can be done in chlorpromazine equivalents. They should also indicate how many patients were screened, which will give readers an idea of the representativeness of their final sample as to the “real world” type of patients they intended to include.
In the statistical section they may want to first state what their hypothesis was, which they are testing. They divide the sample in two groups for some of the assessment instruments. They may want to describe how and why they dichotomized the samples. Their main strategy are multiple correlations, which need to be Bonferroni corrected.
The results are interesting and counter-intuitive Self-reported negative symptoms appear to have the strongest association with the SF-36, while they find no association between their cognitive measure and the SF-36 measures. While the former has certainly been found by many others, the latter runs against findings in much larger studies, such as Mucci et al., (2021) who find social and nonsocial cognition at baseline as primary factors affecting quality of life. Hence, their discussion needs to be a bit more nuanced in view of their limited assessment battery and the limits of a cross-sectional design.
Ref:
Mucci A, Galderisi S, Gibertoni D, Rossi A, Rocca P, Bertolino A, et al. Factors Associated With Real-Life Func-417 tioning in Persons With Schizophrenia in a 4-Year Follow-up Study of the Italian Network for Research on Psychoses. 418 JAMA Psychiatry [Internet]. 2021 May 1.
T
Author Response
Dear Reviewer,
Thank you for your insightful review. It helped us to improve our article significantly. We made changes according to your comments which I wish to share here:
First, you mentioned that our study results might not add new knowledge about HRQoL. As clinicians, we often search for information about quick evaluation tools that could be easily applicable in everyday practice. As mentioned in the article, we usually find information about long and complicated neurocognitive tests and entire battery tests that are difficult to implement in our clinical practice. Therefore, we hope our article will pop up in search results of other clinicians looking for information about quick evaluation tools such as SF-36 MoCA or SNS and show them that these easy-to-use tools yield results that correlate with HRQoL similar to other longer and more detailed tools. Scores of SNS look especially promising because they let us predict HRQoL and therefore have the potential to be used as a well-rounded screening tool.
As you have suggested, we discussed the health-related quality of life (HRQoL) in the introduction in greater detail (see page 2 ‘introduction’). We explained how HRQoL differs from everyday functioning, which refers to how a person perceives his well-being. Moreover, we have expanded the introduction with more information and citations about HRQoL, mentioning that it is inversely correlated with insight and is an independent predictor of relapse.
Also, we have removed the sentence, “It has been accepted that treatment of schizophrenia should aim at improving the HRQoL” from the introduction (see page 2 ‘introduction’). We agree that we overgeneralize, and it has yet to be recognized as the primary treatment goal by any guidelines.
We provided more details about how patients were recruited in the methods section (see page 3 ‘participants’)—all of the authors of this study except prof. S. Dollfus worked in the Psychiatry ward of the Lithuanian University of Health Sciences, which was the study center for our researcher. The authors discussed the inpatients and outpatients with schizophrenia with other psychiatrists who worked in this Psychiatry department and asked them about their patients. Those patients who were deemed not to have active psychotic symptoms by their treating psychiatrist and were in the outpatient setting or on the last day of their inpatient setting were asked to join our study. These patients were then screened, and those that fit the inclusion criteria were included in the study. We did not collect data about other patients. We did not distinguish outpatients and inpatients on their last day of hospitalization as separate patient groups.
You mentioned that we should provide more detail about what type of schizophrenia was included in our study. We have included patients diagnosed with paranoid schizophrenia according to the diagnostic criteria of the International Classification of Diseases 10 (ICD-10) (see page 2 ‘participants’). Therefore, we did not provide any further information about the subtypes of schizophrenia because we did not include them.
We agree with your statement that we have used brief evaluation methods, and MoCA is not a widely recognized tool for cognitive symptoms assessment for schizophrenia. However, we looked for papers that would advise against using MoCA or provided detail about the usage of MoCA for patients with schizophrenia and found that while it performed not as well as neurocognitive batteries, the results acquired with MoCA were still viable. We have cited these sources in the methods section and the discussion (see page 3, ‘procedure and measures’ and page 9, ‘discussion’).
Moreover, as you have suggested, we have improved the methods section with the null hypothesis and Bonferroni correction (see page 4, ‘statistical analysis’). Our null hypothesis was that HRQoL would not correlate with negative symptoms and cognitive deficits of schizophrenia. We had already performed Bonferroni correction in the previous version of the manuscript. However, we should have mentioned that in the methods section.
As you have suggested, we have included a citation from the Mucci et al. paper (see page 9 ‘discussion’), which provides excellent quality data about the relationship between the psychopathology of schizophrenia and the functioning of patients. However, we chose the subjective HRQoL and not the objective functioning of patients as the main interest of this paper. It has been found by other researchers and described in the guidance paper of the EPA that cognitive deficits are linked with worse objective functioning but not with worse subjective HRQoL.
Thank you for your extensive and informative review. We believe you helped us significantly improve our work, and we hope our changes will satisfy you.
Round 2
Reviewer 2 Report
The authors have responded to most of the reviewer's comments. However,iIn the discussion section, the authors need to cite and discuss the findings by Chou et al., who found a very different result as they examine the determinants of subjective HRQoL in a large cohort of schizophrenia patients with tools assessing the same dimensions as the present paper.
Chia-Yeh Chou, Mi-Chia Ma, Tsung-Tsair Yang,
Determinants of subjective health-related quality of life (HRQoL) for patients with schizophrenia,
Schizophrenia Research,
Volume 154, Issues 1–3,
2014,
Pages 83-88,
ISSN 0920-9964,
https://doi.org/10.1016/j.schres.2014.02.011.
Author Response
Dear Reviewer,
Thank you for your further insights into improving our manuscript. We have made corrections according to your comments which I will describe here:
We wrote a separate paragraph in the 'discussion' section (see page 9) where we cited Chou et al. We discussed the different results between Chou et al. and our manuscripts and the reasons behind them. We believe these reasons might lie in different methodologies and questionnaires. Also, we think that often depressive symptoms are masked as secondary negative symptoms in patients with schizophrenia and vice versa, and it is challenging to discern whether we are dealing with depressive or negative symptoms (for example, abolition, blunted affect). Dollfus et al. have found that SNS is an appropriate tool for negative symptoms evaluation regardless of the severity of depressive symptoms present. However, we still believe that further research into the capabilities of self-report diagnostic tools to differentiate between primary and secondary negative symptoms is necessary.
Thank you once again for your comments. They truly helped us enrich our manuscript. We hope that the changes we have made will be sufficient.